# Superhydrophobic and Corrosion Behaviour of PVDF-CeO_2_ Composite Coatings

**DOI:** 10.3390/ma15238674

**Published:** 2022-12-05

**Authors:** Sayed M. Saleh, Fahad M. Alminderej, Adel M. A. Mohamed

**Affiliations:** 1Department of Chemistry, College of Science, Qassim University, Buraidah 51452, Saudi Arabia; 2Chemistry Branch, Department of Science and Mathematics, Faculty of Petroleum and Mining Engineering, Suez University, Suez 43721, Egypt; 3Department of Metallurgical and Materials Engineering, Faculty of Petroleum and Mining Engineering, Suez University, Suez 43512, Egypt

**Keywords:** superhydrophobicity, PVDF-CeO_2_, composite, morphology, corrosion

## Abstract

Composite coatings of polyvinylidene fluoride (PVDF)/CeO_2_ were developed by using the spray approach to explore the wetting and corrosion behaviour of coated materials for applications related to industry. PVDF was combined with different quantities of CeO_2_ nanoparticles followed by spraying onto glass, aluminium, and steel substrates. The sessile droplet method and microscopy studies were used to assess the wetting behaviour and morphology of the coated surfaces, respectively. The corrosion resistance of uncoated substrates coated with PVDF only was compared with those coated with PVDF/CeO_2_ nanoparticles through Tafel polarization techniques. In psi, the force of adhesion was measured between the coating layer and the substrates. The PVDF/CeO_2_-coated steel had a significantly greater water contact angle and lower contact angle hysteresis than coated aluminium and glass substrates, reaching 157 ± 2° and 8 ± 1°, respectively. The corrosion protection efficiency of the superhydrophobic PVDF/CeO_2_ coatings was considerably higher for steel and aluminium when compared with PVDF coatings. The PVDF/CeO_2_ coated substrates had modest adhesion between the coating layer and the substrates, but it was still acceptable. Furthermore, the PVDF/CeO_2_ coatings outperformed PVDF alone in terms of mechanical properties.

## 1. Introduction

Superhydrophobic surfaces (SHCs) have gained a considerable interest because of their outstanding qualities such as self-cleaning, anti-sticking, and anticontamination [1]. Water repellent, self-cleaning, and corrosion-resistant superhydrophobic coatings having a water contact angle (WCA) higher than 150° have piqued the attention in scientific societies [2]. Because the phenomenon of superhydrophobicity is a result of roughness and surface free energy, two ways for producing SHCs have been proposed: chemically changing a substance’s surface of low surface energy, or increasing the target materials’ surface roughness [3,4,5].

Deposition of chemical vapor [6], hydrothermal-dip coating [7], electrochemical deposition [8], sol gel [9], layer-by-layer self-assembly [10], and polymerization by plasma [11] are some of the techniques for producing rough surfaces with a variety of microstructures that have been documented thus far. For developing the superhydrophobic surfaces having a WCA of more than 150°, many processes were implemented, including hydrothermal growth [12], pulsed laser deposition [13], chemical vapor transport and condensation [14], and spray process [15].

The need for a cost-effective and easier approach for rapid manufacturing of superhydrophobic surfaces has increased, since most of the above procedures require expensive components, more time-consuming manufacturing techniques, and the use of dangerous chemicals. The techniques need high priced components, usage of hazardous chemicals, and gradual and slow fabrication techniques; this gained importance of a clear and simple approach for mass-producing superhydrophobic surfaces that does not cost a lot of money or has limitations should be extensively used. 

Spray coating is a cost-effective and straightforward approach for a wide range of applications. It is not bound to a specific type of substrate and can be applied to a large surface area with ease; additionally, it rarely necessitates a more complex and expensive application procedure [16].

Traditionally, polyvinylidene fluoride has been considered to be a hydrophobic material. The thermal and mechanical properties of polyvinylidene fluoride make it suitable for porous polymeric film for use in membrane distillation and filtering [17,18,19]. Even though PVDF has been exposed to numerous methods for enhancing its hydrophobicity to superior levels [20,21], only a few industrial and civil applications of superhydrophobic PVDF-based coatings have been observed. Chaoyi et al. [18], developed a superhydrophobic PVDF coating that possess a WCA and water contact angle hysteresis (WCAH) of 156 ± 1.9 and 2 degrees, respectively, with simple and innovative techniques.

In enhancing the properties of polymers that makes them suitable for particular commercial applications, inorganic nanoparticles such as SiO_2_ [22,23], TiO_2_ [12,24], alumina [25] ZnO [26,27], and ZrO_2_ [28,29] have been used in polymers. A larger fraction of inorganic components is frequently incorporated as a result of these changes. 

Among these inorganic components, the low surface energy, surface roughness, and re-entrant structure of CeO_2_ particles have gained popularity, and all these are vital for attaining hydrophobicity. By combining solid materials such as nickel, cerium, and graphene with the metal oxides such as TiO_2_, SiO_2_, ZnO, and CeO_2_ [30], the surface roughness, which is a parameter of hydrophobicity, can be developed. The toughness and similarity with the organic solvents needed to obtain the PVDF solution have motivated researchers to introduce SiO_2_ instead of CeO_2_ (which is frequently used as an integrated particle) to enhance the abrasion strain and scratch [31,32,33]. In this research, the properties of PVDF/CeO_2_ composite coatings on steel, aluminium, and glass substrates such as wettability, morphology, adhesion, corrosion, and hardness were examined.

## 2. Experimental Work

### 2.1. Materials

PVDF polymer (CH_2_-CF_2_)n and cerium dioxide (CeO_2_) nanoparticles of <100 nm particle size were used as raw materials to synthesize PVDF/CeO_2_ composite coatings. In this study, solvents such as N,N-dimethylformamide (DMF, HCON(CH_3_)_2_, >99%, reagent), stearic acid (CH₃-(CH₂)₁₆-COOH), and hexane (C_6_H_14_) supplied by the AlSAFWA Centre (Cairo, Egypt) were also employed. 

### 2.2. Preparation of the Coatings

Steel, aluminium, and glass substrates were first cleaned using an ultrasonic cleaner in acetone for 8 min, then distilled water for 8 min, and then they were blown dry in a flow of air before being coated. An amount of 5 g of PVDF was dissolved in DMF at 50 °C and stirred at 500 rpm for 1 h to obtain a homogenous solution. A homogenous dispersion of CeO_2_ nanoparticles was generated by dispersing different concentrations of CeO_2_ (1.5, 2, and 2.5 g) in a hexane solution of granules stearic acid and stirring at 500 rpm for 1 h. Obtaining the nanocomposites then required the addition of the CeO_2_ dispersion solution to the PVDF solution. The final solution was created by vigorous mixing with a magnetic stirrer and mild heating. To achieve a flat surface, the final solution was applied using a spray coating procedure on the cleaned substrates. The coated substrates were then dried for 20 min at 80 °C in a drying furnace. The trial conditions are summarised in Table 1.

### 2.3. Characterization Methods

At room temperature, the water contact angle (WCA) and water contact angle hysteresis (WCAH) were determined using the sessile droplet method with an Attension Biolin instrument (Model: Theta Optical Tensiometers, Helsinki, Finland). Readings of WCA and WCAH were taken on the coated substrates’ upper surface using 5 µL distilled water droplets. At least five independent determinations made at different places were combined into one sample. The corrosion behaviour of coated steel and aluminium substrates was investigated by measuring Tafel polarisation curves with a VersaSTAT 3 potentiostat/galvanostat (GAMRY, Munich, Germany). A three-electrode cell comprising a working electrode, a graphite counter electrode, and a saturated calomel reference electrode (SCE) at room temperature in 3.5 wt percent NaCl solution at a scan rate of 2 mVs^−1^ was developed to carry out the corrosion tests. The electrolyte was exposed to 6.25 cm^2^ of the test sample area. The adhesive force between the coating layer and the substrates was measured using a DeFelsko Digital Pull-off Adhesion Tester (Model: PosiTest AT-M, LANGRY, Shandong, China). With a suitable adhesive (epoxy), every sample was attached to a 20 mm diameter dolly. Thereafter, the combination (dolly + sample) was inserted into a tester, and force was applied to divide them. 

The tested samples were sputter coated with Au and vacuum dried prior to morphology analysis. The morphology of the surface for the produced membranes was examined using scanning electron microscopy (SEM Model: JSM-IT200, JOEL, Tokyo, Japan) attached with energy dispersive spectroscopy. 

## 3. Results

### 3.1. Wettability Results

The existence of solid surface energy can be proved by the contact angles since they are the differentiating parameters between solids and liquids. The best condition’s WCAH, which is the difference between the advancing contact angle and receding contact angle, was also observed. Coating a flat solid surfaces with fluorinated methyl groups leads to a maximum WCA of only 120°, which is too low to be considered superhydrophobic [34]. The wettability of the coating affects its anti-corrosion behaviour, as was studied by several researchers [2,19,25,35,36]. CeO_2_ nanoparticles were placed on the solid surface to induce superhydrophobicity. The averages of WCA and WCAH values of PVDF-CeO_2_ composites are shown in Figure 1, Figure 2 and Figure 3.

As shown in Figure 1, the WCA on uncoated steel was about 54 ± 3°, and it was 90 ± 2° when coated with PVDF only, but it climbed to a high of 157 ± 2° upon addition of 2.5 g of CeO_2_ nanoparticles. WCA was 71 ± 3° on uncoated Al substrates and 91 ± 2° when coated with PVDF only, with an extreme of 147 ± 2° when 2.5 g CeO_2_ nanoparticles were added, as revealed in Figure 2. WCA on glass substrates was roughly 51 ± 3° prior to coating, 93 ± 2° after coating with PVDF alone, and 155 ± 3° after adding 2.5 g CeO_2_ nanoparticles, as illustrated in Figure 3. The amount of 2.5 g of CeO_2_ coating on each substrate exhibited the minimum water contact angle hysteresis (WCAH) ranging from 1 ± 0.5° to 5 ± 1 compared with the uncoated specimens or PVDF alone. 

It was suggested to use as an optimal amount the following: For every 100 mL of the final complex solution, 2.5 g of CeO_2_ can be added to enhance the super hydrophobicity of the PVDF. The decrease in the contact angle hysteresis and increase in the water contact angle was observed and demonstrated as follows. As the amount of CeO_2_ added increased, the number of nano-sized asperities generated on the surface increased, which increased the layer roughness. In the space between asperities and the liquid droplets, air was trapped by the nanosized asperities that generated a solid–liquid contact surface in the form of water droplets. This discovery bears some resemblance with the Cassie–Baxter hypothesis [1]. The WCA of air was formerly assumed to be 180°. As stated by Cassie–Baxter’s theory, the Yong–Laplace pressure between the interfaces hinders complete contact between liquid droplets and the solid surface; therefore, the asperities trap air in between, resulting in a stable solid–air–liquid three-phase interface [37].

The WCA on a flat surface (θ) and a rough surface (θ’) composed of a solid surface and air [38] can be drawn from Equation (1).
(1)$$\cos\left(\theta^{\prime }\right)=f_{1}\cos\left(\theta \right)-f_{2}$$

Here, *f*_1_ and *f*_2_ are the solid surface and air ratios in contact with liquid, correspondingly. The value of f_2_ for steel substrates was found to be 0.92, demonstrating that the air was trapped in the roughened hierarchical micro/nanostructures of CeO_2_, which was the main reason for superhydrophobicity in PVDF-based nanocomposites, based on the WCAs of the flat PVDF film (∼90°) and the PVDF nanocomposite (∼157°). As a result, the larger fraction of air trapped would create a higher WCA after the three phases were stabilized. The WCAH and the WCA are both important elements in determining surface hydrophobicity. It was observed that the increase in CeO_2_ concentration in the PVDF polymer to 2.5 g/100 mL of the final complex solution enhanced the nanocomposite’s hydrophobicity when the WCAH of the coating film on steel substrates was decreased from 37 ± 2° in the case of PVDF-only coating to only 5 ± 1° after 2.5 g of CeO_2_ nanoparticles were added (Figure 1),which allows water droplets to easily roll off down the surface.

### 3.2. Corrosion Analysis

The corrosion resistance of PVDF and PVDF/CeO_2_ composite coatings was determined using potentiodynamic polarization curves (Tafel). Figure 4 and Figure 5 show the Tafel polarization curves for steel and Al substrates, respectively, for uncoated substrates; PVDF-coated substrates; and PVDF + 1.5, 2, and 2.5 g CeO_2_-coated substrates at a scan rate of 2 mVs^−1^ in 3.5 wt. percent NaCl solution. Table 2 provides a summary of the Tafel analysis for the polarization curves.

As seen in Table 2, there was a drastic change in the corrosion current density (i_corr_) of PVDF alone-coated steel and (PVDF + 2.5 g CeO_2_)-coated steel from 37.75 in uncoated steel to 3.34 µAcm^−2^ in PVDF alone-coated steel and 3.08 µAcm^−2^ (PVDF + 2.5 g CeO_2_)-coated steel. The i_corr_ of PVDF alone-coated Al and (PVDF + 2.5 g CeO_2_)-coated Al was reduced from 8.26 (uncoated Al) to 1.3 µAcm^−2^ (PVDF alone-coated Al) and 0.2 µAcm^−2^ (PVDF + 2.5 g CeO_2_)-coated Al). However, the protection efficiency was enhanced to 92% with the improved nanocomposite coating (PVDF + 2.5 g CeO_2_) from 91% with the PVDF alone. An enhancement in the protection efficiency of Al from 84% (PVDF only) to 97% (PVDF + 2.5 g CeO_2_) was observed when we improved the nanocomposite coating. Equation (2) gives the protection efficacy of the coatings (*η*) [39].
(2)$$\eta =\frac{i_{1}-i_{2}}{i_{1}}\times 100\%$$

The corrosion current densities of uncoated and coated substrates, respectively, were *i*_1_ and *i*_2_.

The values of polarization resistance, *Rp*, determined using the Stern–Geary equation are also provided in Table 2.
(3)$$R_{p}=\frac{\beta_{a}\beta_{c}}{2.303\left(\beta_{a}+\beta_{c}\right)i_{corr}}$$

*β_a_* and *β_c_* are the anodic and cathodic Tafel slopes, respectively, while the polarization resistance is *R_p_*. The *R_p_* value of the superhydrophobic (PVDF + 2.5 g CeO_2_) was found to be 10 times higher than that of PVDF alone-coated steel, indicating that *R_p_* for the superhydrophobic (PVDF + 2.5 g CeO_2_) nanocomposite coating was about 10 times greater.

The same was true for Al substrates. When the *R_p_* of the superhydrophobic (PVDF + 2.5 g CeO_2_) nanocomposite coating was compared to that of the PVDF alone-coated Al, it is clear that the *R_p_* of the superhydrophobic (PVDF + 2.5 g CeO_2_) nanocomposite coating was 19 times higher.

This means that when coated with (PVDF + 2.5 g CeO_2_) nanocomposite, both steel and aluminium were much less prone to corrosion than when coated with PVDF alone.

At the polymer surface, gaps were found due to trapped air and the inclusion of CeO_2_ nanoparticles on the PVDF polymer, which prevented hostile ions from entering the nanocomposite coating [40]. However, when only PVDF was used, the broad pore with low pore resistance allowed aggressive ions to reach the Al surface.

### 3.3. Adhesion Results

Adhesion is a highly important parameter for the practical application of coatings. Figure 6 depicts the adhesive force of the generated coatings. It is obvious that the adhesion force values for each material of substrate in coated materials followed a similar pattern, with PVDF coatings alone having the highest adherence with the substrates. When CeO_2_ nanoparticles were introduced, this adhesion was diminished. There were high adhesion forces when they were coated with a low level of nanoparticles (1.5 g CeO_2_) rather than coatings with high CeO_2_ concentration.

It is assumed that the high adhesive force of the generated coating surface is mostly due to the effect of cross-linking of PVDF macromolecules through the synthesis of functional groups (C–C or C=C) in the drying process [19]. A decline in the substrate surface’s contact area with the heavy adhesive PVDF was observed by the involved CeO_2_, which made some nanoparticles settle at the interface between the substrate and coating.

The substrate’s surfaces contact area with the heavy adhesive PVDF decreased with the addition of CeO_2_, which allowed some nanoparticles to settle at the interface between the substrate and coating, as shown by the bright areas in the crosscut in Figure 7. In terms of adhesive superhydrophobic properties, coatings of (PVDF + 2.5 g CeO_2_) are preferable, as this combination combines superhydrophobicity with moderate adhesion.

### 3.4. Coatings’ Surface Morphology 

The surface morphology of the generated coatings was investigated using SEM at various magnifications. Figure 8a–c show sponge-like structures in the hydrophobic surface microstructure, showing that the coating layer is predominantly formed of PVDF polymer at high magnification [27,41]. The hydrophobic sample has a uniform surface structure with slight roughness. There are other massive microvoids beyond these formations. Figure 8c shows how EDS was utilised to check the composition of PVDF polymer. According to EDS studies, PVDF material consists mostly of two fluorine (F) and carbon (C) peaks.

As indicated by the bright areas in Figure 9, the addition of CeO_2_ nanoparticles resulted in a considerable change in morphology (a–c). The composite coatings were comprised of CeO_2_ particle aggregates, as seen in Figure 9a. As shown in Figure 9c, the micro–nano scale structured roughness surfaces were made from CeO_2_ particles with a variety of morphologies. This micro–nano scale structure aids the film’s superhydrophobicity, which has a static WCA of 157 ± 2° and a WCAH of 5 ± 1°. The EDS spectrums in Figure 9d reveal the main chemical components of the samples comprising (PVDF + 2.5 g CeO_2_). The PVDF structure had carbon and fluorine, but the integrated CeO_2_ structure contained cerium (Ce) and oxygen (O). EDS elemental analysis of the coated surfaces is shown in Table 3.

## 4. Conclusions

The following are some possible conclusions based on the preceding discussion:PVDF/CeO_2_ composites were successfully made using spray coating processes. This technology has the potential to be employed in wide-scale procedures and to be a cheap solution for industrial applications.The superhydrophobicity of PVDF/CeO_2_ composite coatings is due to the PVDF’s low surface energy and the mixing of hierarchical micro and nanostructures of CeO_2_ incorporated in the polymer.If we raise the concentration of CeO_2_ nanoparticles to 2.5 g/100 mL, the complex solution on the steel surface in the PVDF matrix obtained a considerable increase of WCA from 90 ± 2° to 157 ± 2°, with a fall in WCAH to 5 ± 1°. Both Al and glass substrates produced similar results.By minimizing the size of pores in the composite coating and raising air trapping within the surface’s gaps, CeO_2_ nanoparticles improve the hydrophobicity of PVDF coatings.A significant enhancement in the corrosion resistance of the superhydrophobic PVDF/CeO_2_ composite coatings was observed as 77 times less than the uncoated steel and 177 times lower than the Al substrate. The corrosion rates of the prepared PVDF coating without CeO_2_ nanoparticles were 71 times less than on steel and 44 times lower than on the Al substrates.Although PVDF alone has a stronger adhesive force than (PVDF + 2.5 g CeO_2_) composite coatings, the composite’s adherence is acceptable.

## Figures and Tables

**Figure 1 materials-15-08674-f001:**
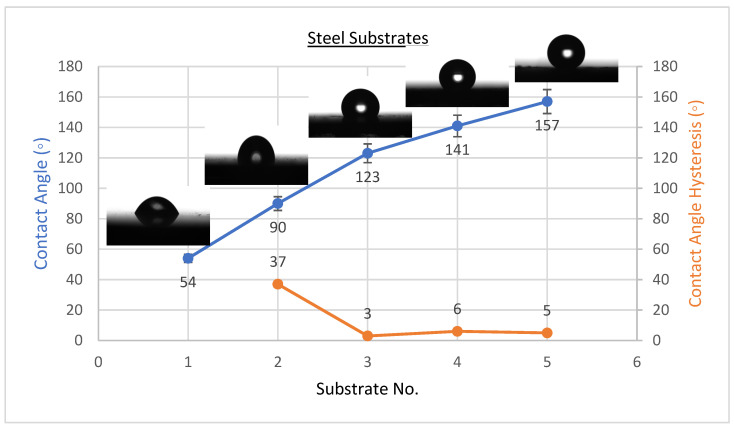
Surface wetting of S1, S2, S3, S4, and S5 samples (steel substrates).

**Figure 2 materials-15-08674-f002:**
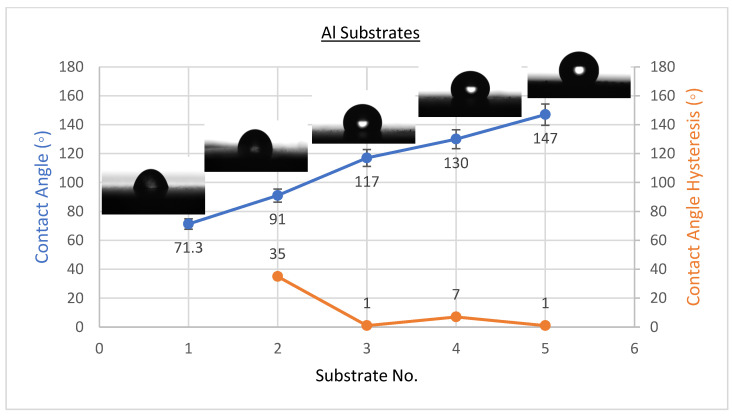
Surface wetting of A1, A2, A3, A4, and A5 samples (Al substrates).

**Figure 3 materials-15-08674-f003:**
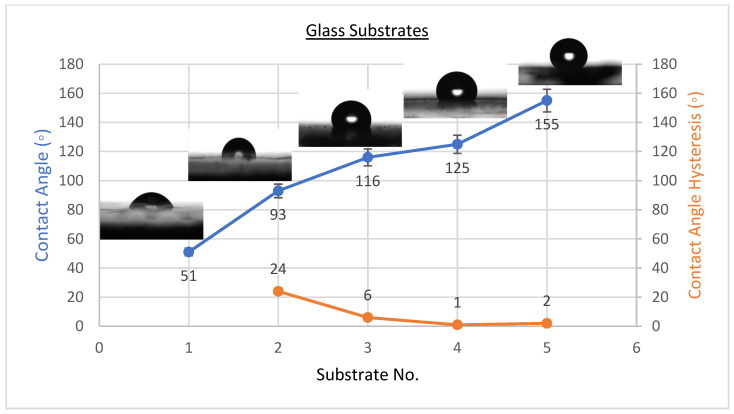
Surface wetting of G1, G2, G3, G4, and G5 samples (glass substrates).

**Figure 4 materials-15-08674-f004:**
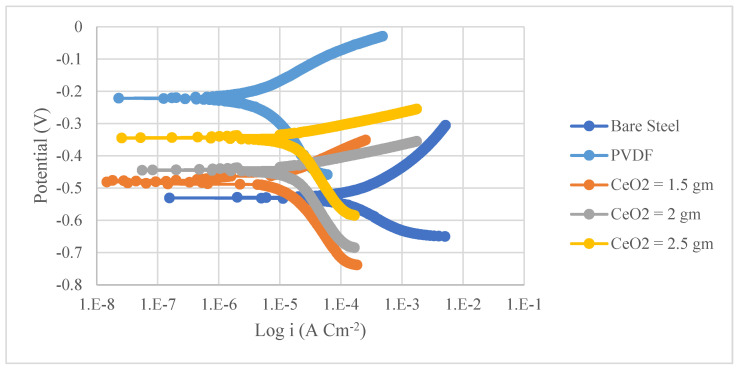
Tafel curves of uncoated steel, PVDF-coated steel, and coated steel with varying CeO_2_ contents of coatings (1.5, 2, and 2.5 g) in a 3.5 percent NaCl solution and a scan rate of 2 mVs^−1^.

**Figure 5 materials-15-08674-f005:**
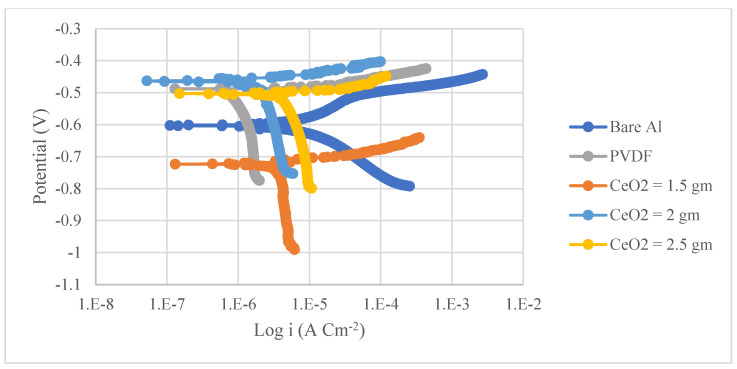
Tafel curves of uncoated Al, PVDF-coated Al, and coated Al with varying CeO_2_ contents of coatings (1.5, 2, and 2.5 g) in a 3.5 percent NaCl solution at a scan rate of 2 mVs^−1^.

**Figure 6 materials-15-08674-f006:**
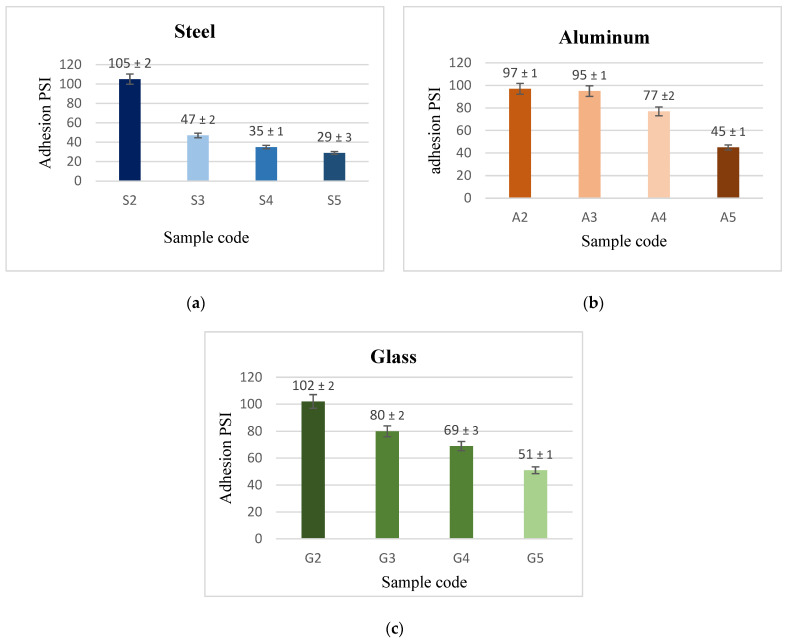
Force of adhesion between (**a**) steel, (**b**) Al, and (**c**) glass substrates, and PVDF alone and PVDF + CeO_2_ coatings.

**Figure 7 materials-15-08674-f007:**
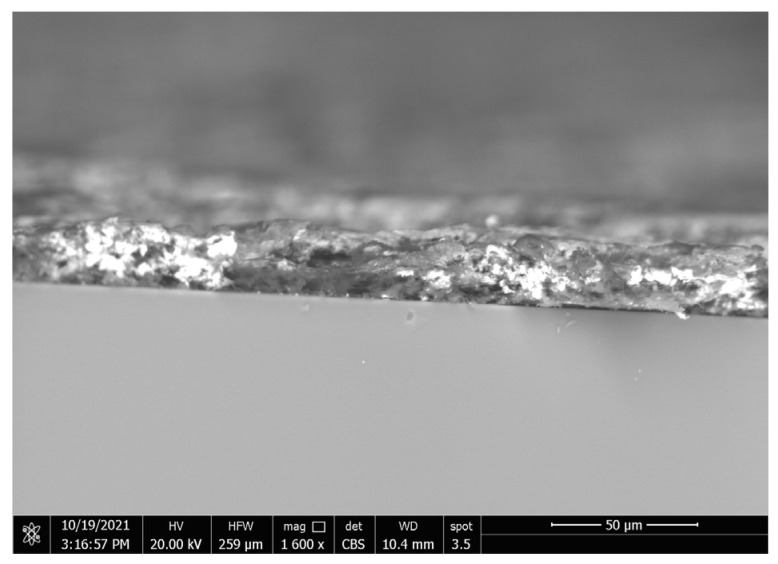
Crosscut SEM images of PVDF composites containing 2.5 g CeO_2_ with nanoparticles on the interface.

**Figure 8 materials-15-08674-f008:**
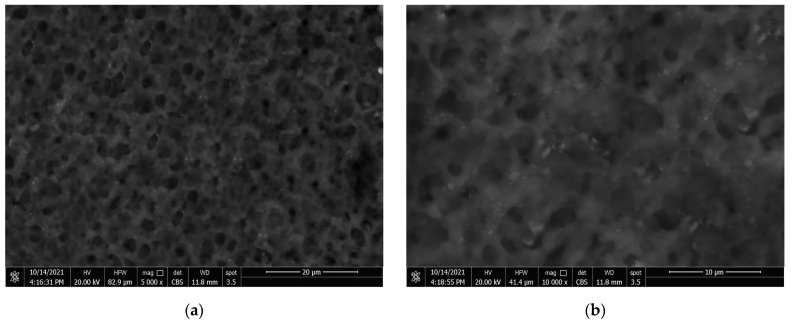
(**a***–***c**) PVDF alone coatings at various magnifications and (**d**) EDS peaks of point 1 in image (**c**).

**Figure 9 materials-15-08674-f009:**
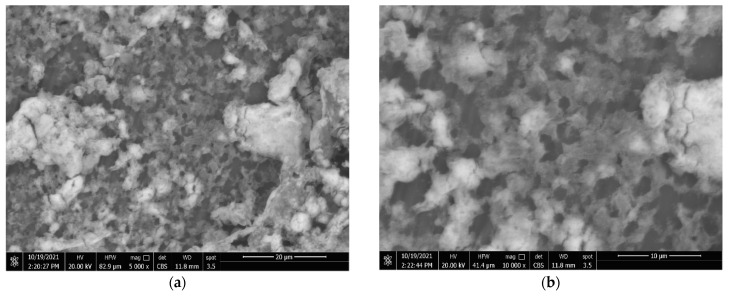
(**a**–**c**) SEM images of PVDF composites containing 2.5 g CeO_2_ at various magnifications and (**d**) EDS peaks of point 1 in image (**c**).

**Table 1 materials-15-08674-t001:** Conditions of the experiments.

Substrate	Coating	Wt/100 mL of Composite Solution	Substrate Code
Steel	Uncoated	-	S1
	PVDF	5.0 g	S2
	CeO_2_	1.5 g	S3
		2.0 g	S4
		2.5 g	S5
Aluminium	Uncoated	-	A1
	PVDF	5.0 g	A2
	CeO_2_	1.5 g	A3
		2.0 g	A4
		2.5 g	A5
Glass	Uncoated	-	G1
	PVDF	5.0 g	G2
	CeO_2_	1.5 g	G3
		2.0 g	G4
		2.5 g	G5

**Table 2 materials-15-08674-t002:** Tafel analysis of uncoated substrates, PVDF alone, and composite coatings (PVDF + 2.5 g CeO_2_) after immersion in a 3.5 percent NaCl aqueous solution; 2 mVs^−1^ was the scan rate.

Steel Substrates	*β_a_* (mV)	*β_c_* (mV)	*R_p_* (K Ω cm^2^)	CR Rate (mpy)	$i_{corr}(µA/{\mathrm{cm}}^{2})$	*η* (%)
Uncoated Steel	117	139	0.7	17	37.75	-
PVDF	73	198	6.9	0.24	3.34	91
PVDF + 2.5 g CeO_2_	538	689	42.6	0.22	3.08	92
**Aluminium Substrates**	***β_a_* (mV)**	***β_c_* (mV)**	** *R_p_* ** **(K Ω cm^2^)**	**CR Rate (mpy)**	$i_{corr}$ ** (µA/cm^2^)**	** *η* ** **(%)**
Uncoated Aluminium	54	122	1.9	3.54	8.26	-
PVDF	93	865	28	0.08	1.3	84
PVDF + 2.5 g CeO_2_	383	658	525	0.02	0.2	97

**Table 3 materials-15-08674-t003:** EDS analysis of surface components on an Al substrate for PVDF alone and (PVDF + 2.5 g CeO_2_) composite coatings.

Element	PVDF Only		PVDF + 2.5 g CeO_2_	
	Mass %	Atom %	Mass %	Atom %
C	55.80	59.35	33.18	53.52
F	44.20	40.65	24.90	25.40
O	-	-	14.25	17.25
Ce	-	-	27.67	3.83
Total	100.00	100.00	100.00	100.00

## Data Availability

Data sharing is not applicable to this article.

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
