# Peer review of "Superhydrophobic and Corrosion Behaviour of PVDF-CeO_2_ Composite Coatings"

_materials, 2022, doi:10.3390/ma15238674_

Round 1
Reviewer 1 Report
This study reported simple preparation of the Superhydrophobic PVDF-CeO2 composite coatings for corrosion protection. This study is suitable for the journal after revision according to the following comments.
1. The quality of the SEM images should be improved. Form the current images, the detailed info cannot be seen.
2. How about corrosion protection performance of the coating in the neutral salt spray test? This is a standard method for evaluation corrosion protective coatings.
3. Some very recent papers about superhydrophobic corrosion protection coatings should be mentioned and compared, such as Chemical Engineering Journal 390 (2020) 124562; Chemical Engineering Journal 390 (2020) 124562.
4. How about mechanical stability of the coating? This should be added, as this is closely related to the lifespan of a superhydrophobic coating.
Author Response
Comment 1: The quality of the SEM images should be improved. Form the current images, the detailed info cannot be seen.
Response to Comment 1: The authors thank the reviewer for his comment. The original-sized images are attached with this file.
- How about corrosion protection performance of the coating in the neutral salt spray test? This is a standard method for evaluation corrosion protective coatings.
Response to Comment 2: yes, I agree with the reviewer that salt spray test is excellent tool to study Unfraternally, the salt spray device is not available in our lab and it is difficult at the moment to do this test in another lab because it will take long time and also for the budget issue, however, we studied the electrochemical behaviors of the coated materials using Potentiodynamic device and obtained good characterization for the corrosion behavior of the samples.
- Some very recent papers about superhydrophobic corrosion protection coatings should be mentioned and compared, such as Chemical Engineering Journal 390 (2020) 124562; Chemical Engineering Journal 390 (2020) 124562.
Response to Comment 3: The authors thank the reviewer for his comment. We have used the suggest references [Ref 35] and another new one [Ref 36]
- How about mechanical stability of the coating? This should be added, as this is closely related to the lifespan of a superhydrophobic coating.
Response to Comment 4: yes, we agreed the reviewer that, the mechanical durability of the developed superhydrophobic coatings is important. In our studied, a quantitative adhesion test was carried out to measure the adhesive properties of the coated materials, which give some info about the mechanical durability of the coated materials.

Reviewer 2 Report
The paper deals with an interesting subjest since stable superhydrophobic surfaces are a matter of high interest for industrial applications. However the paper has various weaknesses:
- In chapter 3.1. sliding angle measurements are mentioned, but in the tables and figures they are never shown and they are not discussed. The sliding angles are probably somehow related to the WCAH but that is never discussed in the paper.
- In line 155 the word maximum is wrong, it should be minimum
- Line 171 and instead of anjd
- Chapter 3.3, line 247: The crosslinking wich is postulated in oder to explain the high adhesion forces would require a dehydroflouorination of PVDF wich would require a high temperature (the drying conditions of 80°C could be sufficient) and the presence of strong bases. This is not necessarily plausible and should at least be backed by spectroscopic evidences.
- chapter 3.3 line 253: The decrease of adhesion force is explained by the settlement of CeO2 nanoparticles in the interface between substrate and coatings. Again there is no experimental evidence for this hypothesis (e.g. crosscuts with SEM investigation).
Author Response
1- In chapter 3.1. sliding angle measurements are mentioned, but in the tables and figures they are never shown and they are not discussed. The sliding angles are probably somehow related to the WCAH but that is never discussed in the paper.
Response to Comment 1: The authors thank the reviewer for his comment and for his correction. Yes, we only measured WCAH not the sliding angle. This mistake has been corrected through the manuscript.
2- In line 155 the word maximum is wrong, it should be minimum
Response to Comment 2: The authors thank the reviewer for his correction. "maximum" word was replaced by "minimum". Please see line 156
3- Line 171 and instead of anjd
Response to Comment 3: The authors thank the reviewer for his correction. "anjd" word was corrected by "and". Please see Line 173
4- Chapter 3.3, line 247: The crosslinking which is postulated in order to explain the high adhesion forces would require a dehydroflouorination of PVDF which would require a high temperature (the drying conditions of 80°C could be sufficient) and the presence of strong bases. This is not necessarily plausible and should at least be backed by spectroscopic evidences.
Response to Comment 4: The authors thank the reviewer for his comment. Some coatings were dried at ambient atmosphere and almost showed similar adhesion results; thus, we tried slight heating just to accelerate the drying process and employ the coating in service. The cross-linking effect has been the nearest hypothesis to explain that behavior of polymer adhesion with and without nanoparticles.
5- chapter 3.3 line 253: The decrease of adhesion force is explained by the settlement of CeO2 nanoparticles in the interface between substrate and coatings. Again, there is no experimental evidence for this hypothesis (e.g. crosscuts with SEM investigation).
Response to Comment 5: The authors thank the reviewer for his comment. A Crosscut SEM image has been added to show the insertion of such CeO2 nanoparticles on the interface between substrate and coatings. The Crosscut image with high quality is attached with the replies to this comment. Please see Figure 7

Round 2
Reviewer 1 Report
The authors did good revision.
Author Response
The Authors would like to thank the reviewer for his feedback and comments for improving the quality of the manuscript
Reviewer 2 Report
Dear Authors, there is only one issue left from my point of view: You made clear in your answer that the coating layers show good adhesion regardless if they are heatet or not. That makes your hypothesis that the adhesion is due to some kind of crosslinking even less likely. That means that the sentence "The high adhesive force of the generated coating's surface is mostly due to the effect of cross-linking of PVDF macromolecules through the synthesis of functional groups (C–C or C=C) in the drying process [19]." (line 249) should be modified, because still there is no evidence for crosslinking. It should be indicated that as an assumption or guess.
Author Response
1- Dear Authors, there is only one issue left from my point of view: You made clear in your answer that the coating layers show good adhesion regardless if they are heated or not. That makes your hypothesis that the adhesion is due to some kind of crosslinking even less likely. That means that the sentence "The high adhesive force of the generated coating's surface is mostly due to the effect of cross-linking of PVDF macromolecules through the synthesis of functional groups (C–C or C=C) in the drying process [19]." (line 249) should be modified, because still there is no evidence for crosslinking. It should be indicated that as an assumption or guess.
Response to Comment 1: The authors thank the reviewer for his comment. We have modified the sentence to be " It is assumed that the high adhesive force of the generated coating's surface is mostly due to the effect of cross-linking of PVDF macromolecules through the synthesis of functional groups (C–C or C=C) in the drying process[19]"
